# Back-to-Africa introductions of *Mycobacterium tuberculosis* as the main cause of tuberculosis in Dar es Salaam, Tanzania

**Michaela Zwyer**[1,2☯], **Liliana K. Rutaihwa**[1,2,3☯], **Etthel Windels**[4,5☯], **Jerry Hella**[3], **Fabrizio Menardo**[6], **Mohamed Sasamalo**[3], **Gregor Sommer**[7], **Lena Schmülling**[8], **Sonia Borrell**[1,2], **Miriam Reinhard**[1,2], **Anna Dötsch**[1,2], **Hellen Hiza**[1,2,3], **Christoph Stritt**[1,2], **George Sikalengo**[3,9], **Lukas Fenner**[10], **Bouke C. De Jong**[11], **Midori Kato-Maeda**[12], **Levan Jugheli**[1,2], **Joel D. Ernst**[13], **Stefan Niemann**[14], **Leila Jeljeli**[14], **Marie Ballif**[10], **Matthias Egger**[10,15,16], **Niaina Rakotosamimanana**[17], **Dorothy Yeboah-Manu**[18], **Prince Asare**[18], **Bijaya Malla**[1,2], **Horng Yunn Dou**[19], **Nicolas Zetola**[20], **Robert J. Wilkinson**[21,22], **Helen Cox**[23], **E Jane Carter**[24], **Joachim Gnokoro**[25], **Marcel Yotebieng**[26], **Eduardo Gotuzzo**[27], **Alash'le Abimiku**[28], **Anchalee Avihingsanon**[29], **Zhi Ming Xu**[5,30], **Jacques Fellay**[5,30,31], **Damien Portevin**[1,2], **Klaus Reither**[2,32], **Tanja Stadler**[4,5], **Sebastien Gagneux**[1,2]*, **Daniela Brites**[1,2]*

**1** Department of Medical Parasitology and Infection Biology, Swiss Tropical and Public Health Institute, Basel, Switzerland, **2** University of Basel, Basel, Switzerland, **3** Department of Intervention and Clinical Trials, Ifakara Health Institute, Bagamoyo, Tanzania, **4** Department of Biosystems Science and Engineering, ETH Zürich, Basel, Switzerland, **5** Swiss Institute of Bioinformatics, Lausanne, Switzerland, **6** Department of Plant and Microbial Biology, University of Zürich, Zürich, Switzerland, **7** Institut für Radiologie und Nuklearmedizin, Hirslanden Klinik St. Anna, Luzern, Switzerland, **8** Klinik für Radiologie und Nuklearmedizin, Department Theragnostik, Universitätsspital Basel, Basel, Switzerland, **9** St. Francis Referral Hospital, Ifakara, Tanzania, **10** Institute for Social and Preventive Medicine, University of Bern, Bern, Switzerland, **11** Unit of Mycobacteriology, Department of Biomedical Sciences, Institute of Tropical Medicine, Antwerp, Belgium, **12** Division of Pulmonary and Critical Care Medicine, University of California, San Francisco, California, United States of America, **13** Division of Experimental Medicine, Department of Medicine, University of California, San Francisco, California, United States, **14** Molecular and Experimental Mycobacteriology, Borstel Research Centre, Borstel, Germany, **15** Centre for Infectious Disease Research and Epidemiology, University of Cape Town, Cape Town, South Africa, **16** Population Health Sciences, Bristol Medical School, University of Bristol, Bristol, United Kingdom, **17** Institut Pasteur de Madagascar, Antananarivo, Madagascar, **18** Bacteriology Department, Noguchi Memorial Institute for Medical Research, College of Health Sciences, University of Ghana, Accra, Ghana, **19** National Institute of Infectious Diseases and Vaccinology, National Health Research Institute, Zhunan, Taiwan, **20** Botswana-UPenn Partnership, University of Pennsylvania, Philadelphia, Pennsylvania, United States of America, **21** Wellcome Center for Infectious Diseases Research in Africa, Cape Town, South Africa, **22** Francis Crick Institute, London, United Kingdom, **23** Institute of Infectious Diseases and Molecular Medicine and the Wellcome Centre for Infectious Disease Research in Africa, University of Cape Town, Cape Town, South Africa, **24** Division of Pulmonary and Critical Care Medicine, Warren Alpert School of Medicine at Brown University, Providence, Rhode Island, United States of America, **25** Centre de Prise en Charge de Recherche et de Formation, Abidjan, Côte d'Ivoire, **26** Division of General Internal Medicine, Department of Medicine, Albert Einstein College of Medicine, New York, New York, United States of America, **27** Instituto de Medicina Tropical "Alexander von Humboldt", Universidad Peruana Cayetano Heredia, Lima, Perú, **28** Institute of Human Virology (IHVN), Abuja, Nigeria, **29** HIV-NAT, Thai Red Cross AIDS Research Centre and Center of Excellence in Tuberculosis, Faculty of Medicine, Chulalongkorn University, Bangkok, Thailand, **30** School of Life Sciences, École Polytechnique Fédérale de Lausanne, Lausanne, Switzerland, **31** Precision Medicine Unit, Lausanne University Hospital and University of Lausanne, Lausanne, Switzerland, **32** Department of Medicine, Swiss Tropical and Public Health Institute, Basel, Switzerland

☯ These authors contributed equally to this work.

\* sebastien.gagneux@swisstph.ch (SG); d.brites@swisstph.ch (DB)

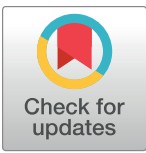

**Data Availability Statement:** The newly sequenced and unpublished WGS data can be found under the bioproject PRJEB49562 on ENA. Xml files for the different phylodynamic analyses are provided as

extended data (https://github.com/dbrites/TB-DAR-Mtb).

**Funding:** This work was supported by the Swiss National Science Foundation (https://www.snf.ch; Grant No: CRSII5_177163, 310030_188888) and the European Research Council (https://erc.europa.eu/; Grant No: 883582). RJW is supported by the Francis Crick Institute which receives funding from Wellcome (FC0010218), Cancer Research UK (FC0010218), and the Medical Research Council (FC0010218) and he also receives support from Welcome (203135). IeDEA is supported by the US National Institutes of Health, National Institute of Allergy and Infectious Diseases, the Eunice Kennedy Shriver National Institute of Child Health and Human Development, the National Cancer Institute, the National Institute of Mental Health, the National Institute on Drug Abuse, the National Heart, Lung, and Blood Institute, the National Institute on Alcohol Abuse and Alcoholism, the National Institute of Diabetes and Digestive and Kidney Diseases, the Fogarty International Center, and the National Library of Medicine: Asia-Pacific, U01AI069907; CCASAnet, U01AI069923; Central Africa, U01AI096299; East Africa, U01AI069911; NA-ACCORD, U01AI069918; Southern Africa, U01AI069924; West Africa, U01AI069919 and the Swiss National Science Foundation (Ggrant No: number 320030_153442 and 189498). The funders had no role in study design, data collection and analysis, decision to publish, or preparation of the manuscript.

**Competing interests:** The authors have declared that no competing interests exist.

# Abstract

In settings with high tuberculosis (TB) endemicity, distinct genotypes of the *Mycobacterium tuberculosis* complex (MTBC) often differ in prevalence. However, the factors leading to these differences remain poorly understood. Here we studied the MTBC population in Dar es Salaam, Tanzania over a six-year period, using 1,082 unique patient-derived MTBC whole-genome sequences (WGS) and associated clinical data. We show that the TB epidemic in Dar es Salaam is dominated by multiple MTBC genotypes introduced to Tanzania from different parts of the world during the last 300 years. The most common MTBC genotypes deriving from these introductions exhibited differences in transmission rates and in the duration of the infectious period, but little differences in overall fitness, as measured by the effective reproductive number. Moreover, measures of disease severity and bacterial load indicated no differences in virulence between these genotypes during active TB. Instead, the combination of an early introduction and a high transmission rate accounted for the high prevalence of L3.1.1, the most dominant MTBC genotype in this setting. Yet, a longer co-existence with the host population did not always result in a higher transmission rate, suggesting that distinct life-history traits have evolved in the different MTBC genotypes. Taken together, our results point to bacterial factors as important determinants of the TB epidemic in Dar es Salaam.

## Author summary

Tuberculosis (TB) is among the deadliest human infectious diseases caused by one single agent, *Mycobacterium tuberculosis* (Mtb). The origins of Mtb have been traced to East Africa millennia ago, where it likely became adapted to infect and transmit in humans. Here, we show that in Dar es Salaam, Tanzania, an East African setting with a high burden of TB, infections are caused by distinct Mtb genotypes introduced in recent evolutionary times from different parts of the world. These genotypes differed in traits important to Mtb transmission; while some Mtb genotypes transmitted more efficiently during a given period of time, patients infected by other genotypes remained infectious for longer. These traits evolved independently in the different Mtb genotypes and could not be explained by the time of co-existence between the host population and the pathogen. This suggests that bacterial factors are important determinants of the TB epidemic. More generally, we demonstrate that distinct pathogenic life history characteristics can co-exist in one host population.

## Introduction

Tuberculosis (TB) is an airborne disease caused by members of the *Mycobacterium tuberculosis* Complex (MTBC) and is among the leading causes of human death due to a single infectious agent. The COVID-19 pandemic has negatively affected TB case notification, treatment, and the number of TB deaths [1]. While the TB death toll had been decreasing each year since 2005, it is increasing again since 2020, with an estimated 1.6 million deaths in 2021 [1]. Of these, 12% occurred in HIV co-infected patients [1], highlighting HIV infection as risk factor for TB [2].

Within the MTBC, nine human-adapted phylogenetic lineages have been described to date; lineage 1 (L1) to L9. Even though the members of the MTBC are highly clonal, and individual strains share more than 99% DNA sequence similarity [3], clinical strains differ in their phenotypes [4]. For example, MTBC strains have been reported to exhibit variable growth rates in macrophages, differences in the host immune responses elicited, differences in gene expression, as well as differences in transmissibility [4–9].

The MTBC as a whole is hypothesized to have originated in East Africa [10,11], which is supported by most MTBC genetic diversity being found in that part of the world [12]. It has been further hypothesized that at some point during its evolution, the MTBC spread out of Africa and diversified in different regions around the world [13,14]. Throughout the last 600 years, lineages that evolved outside of Africa were brought back to Africa following waves of exploration, trade and conquest [15–18]. Despite centuries of trade and migration, many MTBC genotypes remain highly restricted to specific geographical regions where, in some cases, they have also been associated with particular human ethnicities. For example, L1 occurs mainly along the rim of the Indian Ocean, L5 is restricted to West Africa and has been associated with the Ewe ethnicity in Ghana [19], and the Beijing sublineage of L2 has been linked to the Hui ethnicity in China [20]. By contrast, L4 occurs worldwide, although some L4 sublineages are restricted to certain geographical regions like L4.6.1, which is strongly linked to Uganda and some neighboring countries, or L4.5 that mainly occurs in Asian countries [15]. Frequencies of lineages and sublineages can differ markedly between neighboring countries [21], and even within a single country [22]. Such patterns of phylogeographical associations are compatible with the notion that MTBC genotypes might be locally adapted to specific human populations. This notion is supported by the observation that these patterns remain stable in cosmopolitan settings [23–25]. However, alternative explanations for the phylogeographical associations of particular MTBC genotypes can be invoked, such as founder effects. Based on the current knowledge, however, why some MTBC lineages or sublineages predominate in a particular geographical region remains largely elusive. Microbial, environmental and host factors, as well as human migrations are likely at play [9,17,18,26–29]. It has also been suggested that the genetic make-up of particular bacterial populations may influence the spread of certain MTBC groups, such as L2 in Asia [30]. However, how the interplay between these forces shapes the composition of local MTBC populations in high-burden TB settings is poorly understood.

Here we investigated the evolutionary history, the epidemiological characteristics and the clinical phenotypes associated with the TB epidemic in Dar es Salaam, Tanzania, a TB high-burden country in East Africa. We used whole genome sequences (WGS) from MTBC isolates from TB patients recruited at a TB clinic in Dar es Salaam during six years, together with their clinical data. We provide evidence that the current MTBC population structure mainly comprises MTBC genotypes that were introduced from outside of Africa, but at different times during the last 300 years, and that these genotypes differ in their life history traits and associated epidemiological characteristics.

## Results

### The TB epidemic in Dar es Salaam—Patient and pathogen characteristics

We prospectively recruited 1,734 GeneXpert-positive adult TB patients at the TB clinic in the Temeke District of Dar es Salaam, Tanzania, between November 2013 and August 2019 (Table 1). Dar es Salaam has the highest TB notification rate in Tanzania [31]. Temeke is one of three districts in Dar es Salaam contributing to about a third of all TB notifications in the city (personal communication by Jerry Hella), with a TB notification rate of 297/100,000

**Table 1. Clinical and sociodemographic characteristics of patients recruited.** The tribes named are those with at least 70 members among our patient population.

| label | Total N (%) | Missing N (%) | levels | all |
|---|---|---|---|---|
| Total N (%) | | | | 1734 (100.0) |
| Sex | 1734 (100.0) | 0 (0.0) | Female (%) | 512 (29.5) |
| | | | Male (%) | 1222 (70.5) |
| Age | 1734 (100.0) | 0 (0.0) | Mean (SD) | 34.9 (10.8) |
| Smoker | 1729 (99.7) | 5 (0.3) | No (%) | 1331 (77.0) |
| | | | Yes (%) | 398 (23.0) |
| Xray-score | 1137 (65.6) | 597 (34.4) | Mean (SD) | 42.3 (29.9) |
| TB-score | 1734 (100.0) | 0 (0.0) | Mean (SD) | 5.0 (1.6) |
| BMI | 1734 (100.0) | 0 (0.0) | Normal (%) | 740 (43) |
| | | | Obese (%) | 16 (1) |
| | | | Overweight (%) | 66 (4) |
| | | | Underweight (%) | 912 (53) |
| HIV status | 1716 (99.0) | 18 (1.0) | Infected (%) | 365 (21.3) |
| Tribes | 1734 (100) | 0 (0.0) | Makonde (%) | 134 (8) |
| | | | Ndenereko (%) | 268 (15) |
| | | | Zaramo (%) | 194 (11) |
| | | | Chaga (%) | 82 (5) |
| | | | Mwera (%) | 103 (6) |
| | | | Other (%) | 1138 (66) |

population in 2019 [32]. The number of patients recruited per year varied between 195 and 364 (not considering 2013 and 2019, which were only partially sampled). Males were overrepresented among patients (71%), and HIV coinfection was more prevalent among female TB patients (33% vs. 16% in males), which was consistent with the generally higher prevalence of HIV in women in Dar es Salaam (6% vs. 2% in males) [33]. Chest X-ray scores were lower in HIV-co-infected TB patients compared to HIV-negative TB patients with a mean of 29.4 (SD: 29.1) versus 45.8 (SD: 29.2), respectively (p-value < 0.001, ANOVA), reflecting atypical lung pathologies in HIV co-infected patients. From the 1,734 patients recruited, we obtained bacterial DNA from 1,155 unique patient samples (66%) and a final number of 1,082 MTBC WGS that passed quality filters (S1 Fig). Patients without a bacterial WGS available (n = 652, 38%), had a significantly lower chest X-ray score than patients with a bacterial WGS available (p = 0.001, ANOVA), suggesting that viable bacteria are more likely to appear in sputum from patients with increased lung damage. There were no other substantial differences in the sociodemographic and clinical characteristics between patients with and without bacterial WGS data (S1 Table).

The phylogenetic analysis of the 1,082 MTBC genomes revealed that four of the nine known human-adapted MTBC lineages circulate in Dar es Salaam (Fig 1). L3 was the most prevalent with 47% of all isolates, followed by L4 (31%), L1 (14%), and L2 (8%). The lineage proportions fluctuated over the years (S2 Fig); but there was no marked trend over time. The most common sublineages were L3.1.1 (41%), L4.3.4 (15%), L1.1.2 (11%), and L2.2.1 (8%). Patient characteristics did not differ statistically across the four lineages nor across the main sublineages (S2 and S5 Tables).

When screening our MTBC genomes for drug resistance mutations, we found that only 55 (5%) contained at least one mutation conferring resistance to first-line drugs (S3 Table), and only two (0.2%) were multidrug-resistant. The proportion of strains that were resistant to at least one first-line drug differed between lineages, with 10% in L4 (N = 34), 4% in L1 (N = 6),

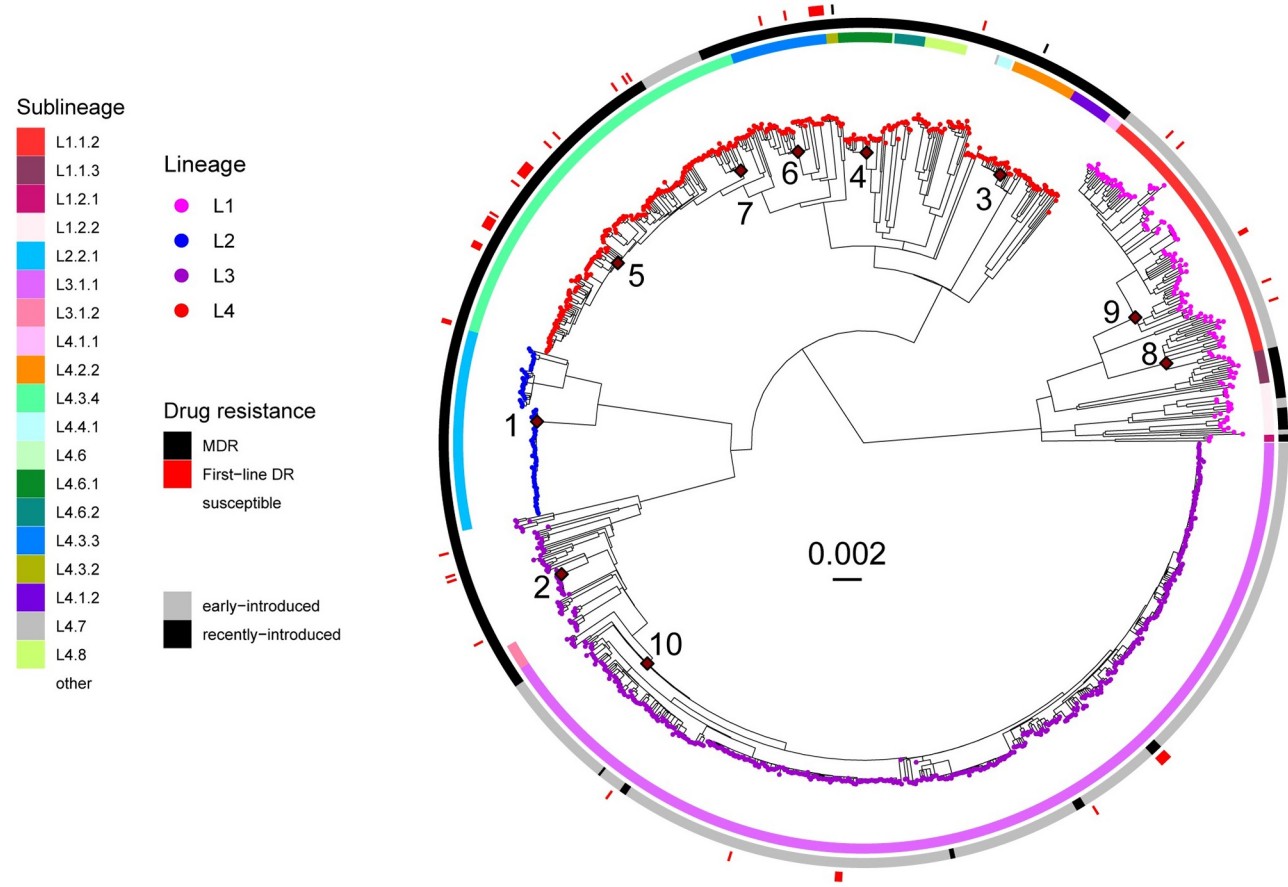

**Fig 1. Phylogeny of 1082 Mtb genomes sampled from 2013–2019 in Dar es Salaam.** The tree is rooted with a *M. canettii* strain (SAMN00102920) and the scale bar indicates substitutions per site. Tips are colored according to the MTBC lineage and the innermost heatmap indicates MTBC sublineages according to [47]. The second heatmap indicates whether a strain is considered as recently-introduced or early-introduced based on a threshold of 0.2 for the relative age (See methods). The outermost heatmap indicates the genotypic drug resistance profiles for most commonly used drugs in Dar es Salaam (See methods). Only mutations giving rise to first-line drugs are considered. The MTBC introductions into Tanzania leading to most cases in our cohort are labelled from 1–10.

3% in L3 (N = 15), and none in L2 (S4 Table). Testing for associations between first-line drug resistance and different bacterial and patient characteristics showed that L4 was associated with resistance to first-line drugs (logistic regression corrected for age, sex, smoking, and HIV status, p < 0.001, S4 Table).

In summary, based on the 1,082 *Mtb* genomes analyzed, we found that the TB epidemic in Dar es Salaam is caused by L1, L2, L3, and L4, with L3 being particularly dominant. Patient and bacterial characteristics were similar between MTBC lineages and sublineages circulating in Dar es Salaam, apart from resistance to first-line anti-TB drugs, which was associated with L4. However, the overall prevalence of drug resistance to first-line drugs in Dar es Salaam was low in comparison to other African cities with a high burden of TB.

## Geographic and temporal origins of the Dar es Salaam TB epidemic

It has been hypothesized that the MTBC originated in Africa [3,23], and that subsequent migrations out of and back to Africa have shaped the genetic landscape underlying the TB epidemic on the continent [13,16–18]. Dar es Salaam had many trade links in the past, through

the Indian Ocean with Central- and South Asia and later with Europe, presumably explaining the high genetic diversity of the MTBC found in our sample set. We thus explored in more detail how migration might have shaped the MTBC diversity in Dar es Salaam by inferring the geographical and temporal origins of the MTBC strains circulating in the city.

For this, we put the MTBC genomes from Dar es Salaam into a global context by assessing their phylogenetic placement within lineage-specific representative reference sets of MTBC genomes gathered world-wide (S7 Fig and S10 Table). For each lineage, separate phylogeographic patterns were inferred using PastML [34] (S3–S6 Figs). Most L1 and L3 strains in Dar es Salaam were predicted to be introduced from South- or Central Asia. For L2, most strains were introduced from East Asia and a few possibly directly from other African regions after being introduced from East Asia. L4 strains had many different geographic origins but predominately were inferred to be introduced from South America. Given the history of European colonization, most likely L4 strains were introduced by Europeans to both Africa and South America as also inferred by others [17]. However, a direct connection between African and European strains is not possible to infer as the latter have disappeared with the decline of TB in Europe [17]. The exception was L4.6 whose ancestors seemed to have originated in Central Africa. These findings could be affected by missing data due to extinctions of local populations or due to incomplete sampling. However, the general patterns are in agreement with previous studies quantifying MTBC dispersal from and towards Africa [16–18,27]. As for the most likely geographic ranges inferred for the MRCAs of L1 to L4 globally, our inferences point to South- or Central Asia for L1 and L3, Eastern- or Southeastern Asia for L2, and Eastern Asia, Southern Asia, Eastern Europe, or Western Africa for L4. These findings are also in agreement with previous studies [18,27,35,36] except for L4, for which the geographic location of the MRCA has been predicted to Europe or Eastern Africa [17,27]. In summary, even though East Africa is the most probable origin of the MTBC as a whole [14], the strains sampled in our cohort were most likely introduced into Tanzania from different parts of the world.

We next determined the MTBC introductions into Tanzania that spread more successfully within Dar es Salaam, as well as the timing of these introductions. We then used the latter as an approximation for the time that these different MTBC populations evolved with this host population. We reasoned that the most successful introductions were those that left more descendants, and which therefore were more prevalent in our patient population. We identified introductions into Tanzania that led to at least 12 sampled cases within our patient cohort (Fig 1). Based on dated trees generated for each lineage separately, we dated each introduction according to estimated lineage-specific substitution rates from our data and from other publications (see methods for further details, S9 Table).

In total, we identified ten independent introductions represented by at least 12 monophyletic strains leading to TB cases in our cohort. These strains have thus evolved by infecting and transmitting within this Tanzanian population for several generations. The most successful introduction involved sublineage L3.1.1 (Introduction 10) that came from South- or Central Asia an estimated 312 years ago (max: 899, min: 273) and accounted for 38.9% of all current cases (Figs 1, 2 and S5). The second most successful introduction, also from South- or Central Asia, occurred an estimated 256 years ago (max: 763, min: 165) within L1.1.2 (Introduction 9) and contributed 8.3% of all current cases (Figs 1, 2 and S3). From the same geographic region and of similar estimated age, a second introduction (Introduction 8) occurred within L1.1.2, 237 years ago (max: 697, min: 151) but accounted for fewer current cases (1.9%) (Figs 1, 2 and S3). More recently, an estimated 57 years ago (Introduction 2, max: 157, min: 50), an unclassified group of L3 strains was introduced, accounting for 1.3% of all infections in our cohort (Figs 1, 2 and S5).

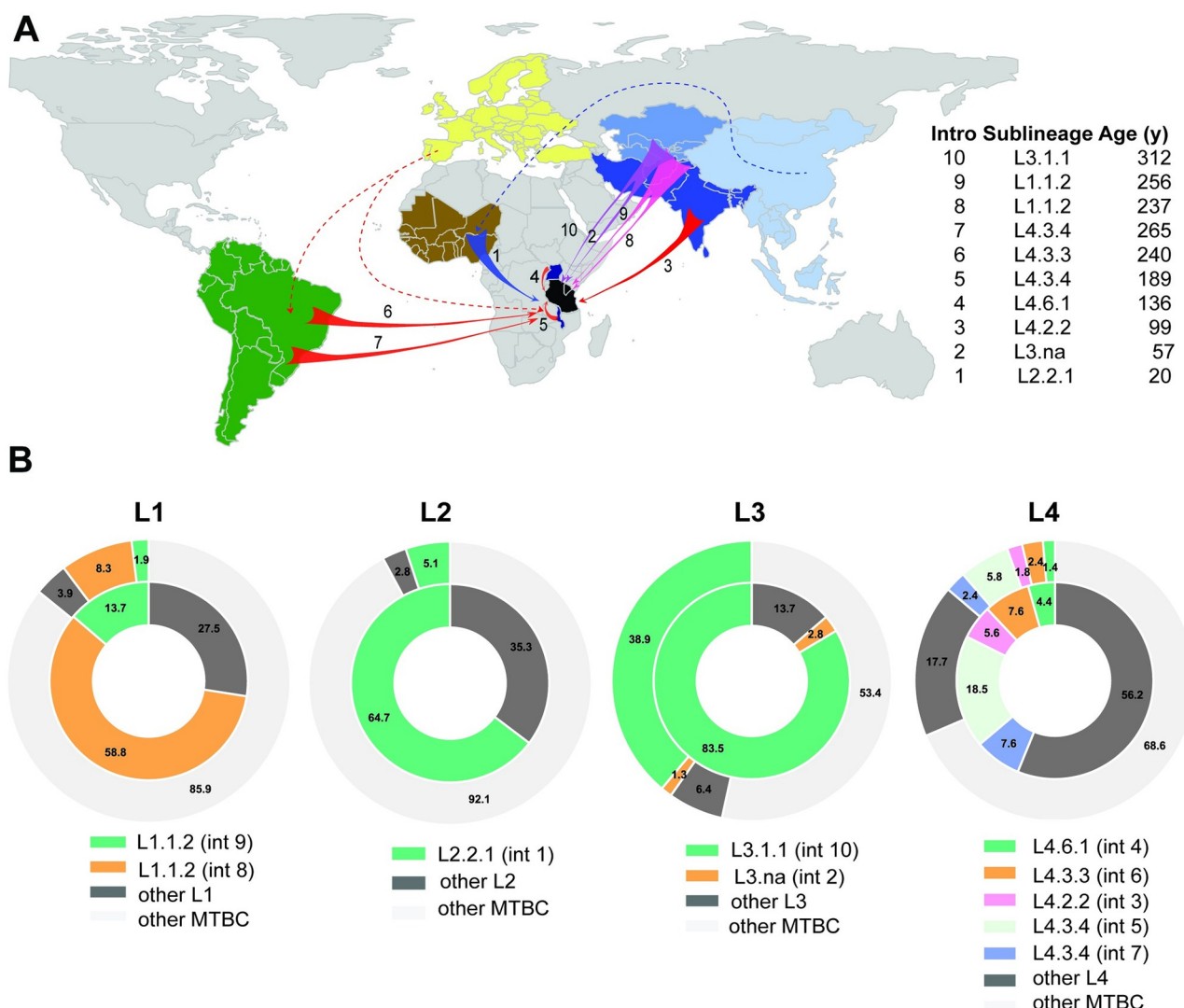

**Fig 2. The genotypes in Dar es Salaam resulting from introductions 1–10.** A—Geographic origin of the 10 introductions into Tanzania that led to most cases in our cohort. Introductions are labelled as in Fig 1 and are represented by colored arrows according to the lineage. Regions or countries identified as the origin of a successful introduction are colored (dark green: South America; brown: West Africa; dark blue: Malawi and Uganda; black: Tanzania; shades of blue: South, Central and East Asia; yellow: Europe). The age of the introductions were obtained from substitution rates inferred from our dataset except for L1 (See methods and S8 Table for details). The map was created with the R package rworldmap [109] and the shapefile for the map can be found under the following link: https://www.naturalearthdata.com/http/www.naturalearthdata.com/download/110m/cultural/ne_110m_admin_0_countries.zip. B—Prevalence of the most prevalent genotypes within each lineage (inner circle) and across all lineages (outer circle).

L4 had the highest number of independent introductions that spread successfully in Dar es Salaam (Introductions 3 to 7). The most successful introduction was that of L4.3.4, inferred to be introduced from what is today Malawi (Introduction 5), an estimated 189 years ago (max: 350, min: 189) and contributing to 5.8% of all cases (Figs 1, 2 and S6). Other subgroups within L4.3, also known as Latin American Mediterranean family (LAM), L4.3.3 (Introduction 6) and L4.3.4 (Introduction 7), were inferred to be introduced from South America; 240 (max: 445, min: 240) and 265 (max: 493, min: 265) years ago, respectively. The different independent introductions of the different groups within sublineage L4.3 could have also occurred directly from Europe to both South America and Africa, reflecting the close links between the three

continents during the colonial period. The remaining significant introductions involved L4.6.1 (Introduction 4) from Uganda an estimated 136 years ago (max: 253, min: 136) and L4.2.2 (Introduction 3) from Southern Asia, 99 years ago (max: 182, min: 99). Each of the latter introductions accounted for 1.4–2.4% of all infections (Figs 1, 2 and S6). Finally, for L2, we found only one successful introduction to Dar es Salaam of a group within sublineage L2.2.1, which we inferred to be introduced from Asia via West Africa around 20 years ago, accounting for 5.1% of all current cases (Introduction 1, Figs 1, 2 and S4). An alternative scenario would be that L2.2.1 was introduced to West Africa and to East Africa directly from Asia, but its closest related Asian strains were not sampled (S4 Fig). The strains belonging to these 12 most successful introductions accounted for 58% of all strains circulating in the city. The remaining strains belonged to many other genotypes within the four main lineages, which individually did not expand as successfully in our cohort, but which together accounted for 42% of all infections (S3–S6 Figs).

Since it is known that phylogeographic reconstructions can be affected by sampling bias [37,38], we carried out a sensitivity analysis and repeated our geographical inferences 10 times with down-sampled datasets for L1 and L3. The main introductions within L1 and L3 remained the same and the timing changed only marginally (S10–S11 Figs). The phylogenetic reconstructions resulting from the down-sampling can be found in the extended data (https://github.com/dbrites/TB-DAR-Mtb).

In summary, strains belonging to the four MTBC lineages L1, L2, L3, and L4 were introduced into Tanzania on multiple occasions between an estimated 20 and 312 years ago from diverse regions of the world. Following their introduction, these strains diversified in Tanzania, and some introductions became the source of many TB cases in our cohort while others were not as successful.

## Early and recently introduced strains do not differ in virulence

We hypothesized that strains that have been circulating for a longer period could be better adapted to the host population residing in Dar es Salaam compared to strains introduced more recently. Due to the high uncertainties in estimating substitution rates [39], we used the relative ages of introduction instead of the absolute ages. A strain from Dar es Salaam was defined as "early-introduced" if the most basal node having Tanzania as the inferred ancestral range had a relative age greater than 0.2 relative to the age of the most recent common ancestor (MRCA) of the respective lineage the strain belonged to. Thus, at least 20% of a genotype's evolutionary history had to have happened in Tanzania for the descendants of a particular introduction to be considered "early-introduced". Conversely, all the descendants of introductions dated to have occurred at less than 0.2 of the total age of the tree, were considered "recently-introduced".

We found that the TB epidemic in Dar es Salaam was driven to almost equal parts by early-introduced (52%) and recently-introduced strains (48%). However, there were marked differences between lineages: while for L1 and L3 most strains were classified as early-introduced (83.5% and 78.4%, respectively), most strains in L4 (92.4%) and all in L2 were classified as recently-introduced. We hypothesized that early-introduced strains could be locally adapted to the patient population in Dar es Salaam, which might reflect in differences in virulence between early-introduced and recently-introduced strains. We defined virulence as the degree of harm caused to the patient, and used as proxies for virulence the following three measures of disease severity: TB score, chest X-ray score, and bacterial load. We found that whether a strain was early-introduced or recently-introduced did not influence the disease severity in the infected patients based on these three proxies (S6 and S7 Tables). Applying different

introduction age thresholds for defining "early- as opposed to late-introduced" did not reveal any relevant differences either.

In summary, we found that the TB epidemic in Dar es Salaam is driven both by early-introduced and recently-introduced strains in similar proportions. Despite the fact that some strains were introduced earlier, and thus had more time to evolve with this particular host population, we did not observe any effect on virulence based on the three measures of disease severity considered here.

## High prevalence is not only a consequence of early introduction

The high prevalence of certain MTBC genotypes could simply reflect an earlier introduction into Dar es Salaam, assuming that the host population in Dar es Salaam was equally susceptible to all MTBC genotypes introduced, and that life-history traits affecting infectiousness and transmission of those MTBC genotypes did not differ at the time of introduction. If time since introduction would be a strong determinant of the current prevalence in the population, we would expect a positive correlation between the number of strains descending from a particular introduction and the relative age of this introduction. We tested this, considering the ten most important introductions and found a moderate correlation, which did not reach statistical significance (Fig 3). This suggests that time since introduction could determine to some extent the prevalence of the most common MTBC groups in Dar es Salaam. However, this effect was mainly driven by the most common genotype descending from "Introduction 10" within L3.1.1 (Fig 3), which was introduced earlier than others (i.e. estimated relative age of 0.33 or 312 years ago, Figs 2C and 3). However, Introductions 9 and 8 of L1.1.2 (Fig 3) (estimated relative age of 0.32 or 256 years ago and 0.30 or 237 years ago, respectively) happened not long after, and yet, the number of resultant TB cases was similar to those of more recent introductions. Also, given the recent introduction of L2.2.1 (Introduction 1) into Dar es

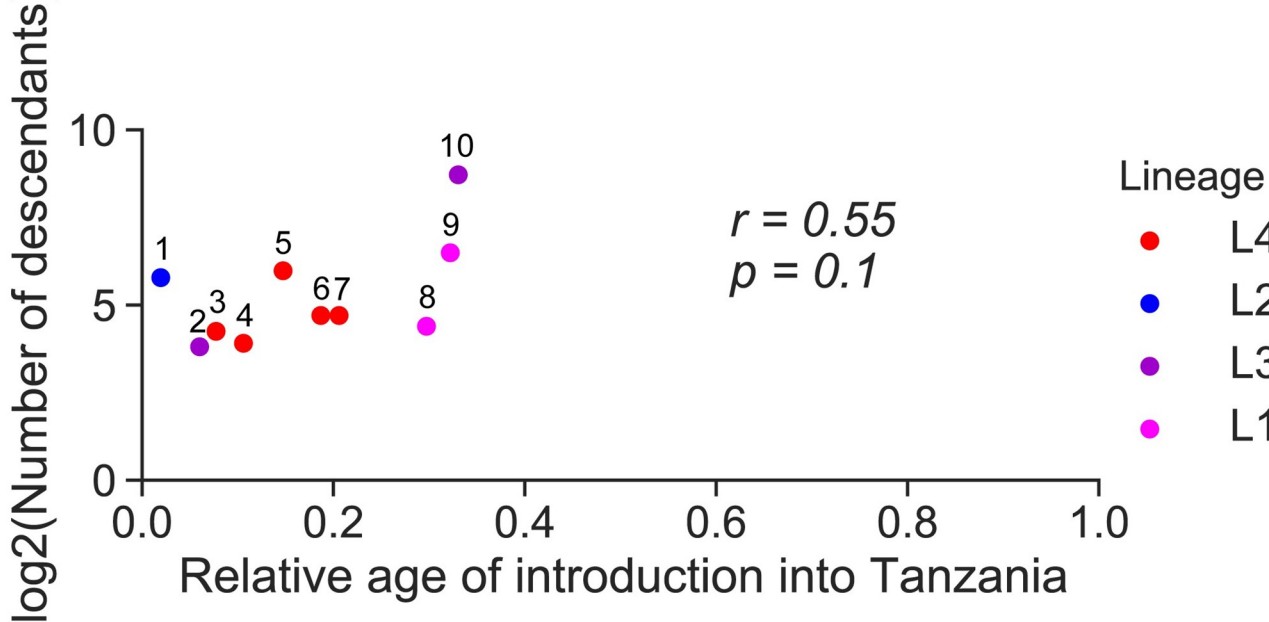

**Fig 3. Number of descendants resulting from the ten most successful introductions and the relative age of the latter.** The number above each introduction corresponds to the numbers in Figs 1 and 2. The pearson correlation coefficient (r) and p-value (p) were calculated.

Salaam (an estimated 20 years ago), its current prevalence was surprisingly high (Fig 3). These findings suggest that there are other factors, in addition to the time of co-existence between host and pathogen populations that determine the prevalence of the different MTBC genotypes in Dar es Salaam.

## Differences in transmission between genotypes

Our results suggested that the different MTBC genotypes infecting our patient cohort did not differ in virulence at the stage of active disease. However, they could differ in other life-history traits affecting transmission. Focusing on the four main monophyletic groups circulating in our cohort, which belonged to L3.1.1, L4.3.4, L1.1.2 and L2.2.1, we investigated whether differences in transmission could also account for the observed differences in prevalence. For this, we analyzed recent MTBC transmission within our cohort using as proxies different measures of clustering based on pair-wise distances and age thresholds, as well as terminal branch lengths.

Thresholds of five to 12 SNPs have previously been shown to detect clusters linked to recent transmission in the MTBC [40,41]. However, because clustering based on SNP thresholds does not consider variable substitution rates across the different MTBC lineages [39], the same SNP threshold might reflect different properties in different genetic backgrounds. To account for this, we also defined clusters based solely on time to the most recent ancestor, using an increasing age threshold from five to 20 years, for each of the four most successful introductions. All genomes that had a common ancestor dated at one of the thresholds from five to 20 years ago, based on the estimated substitution rate for each lineage, were considered to belong to a transmission cluster.

Using the identified clusters, we calculated the secondary case rate ratios comparing secondary case rates of L2.2.1, L4.3.4, and L1.1.2 to that of L3.1.1 representatives in our population. Clustering based on both SNP and age thresholds revealed that L2.2.1 (Introduction 1) had the highest secondary case rates (Fig 4A and 4B) and that the strains from L1.1.2 (Introduction 9) and from L4.3.4 (Introduction 5) had lower secondary rate ratios than strains from L3.1.1 (Introduction 10), in particular when considering higher SNP thresholds and

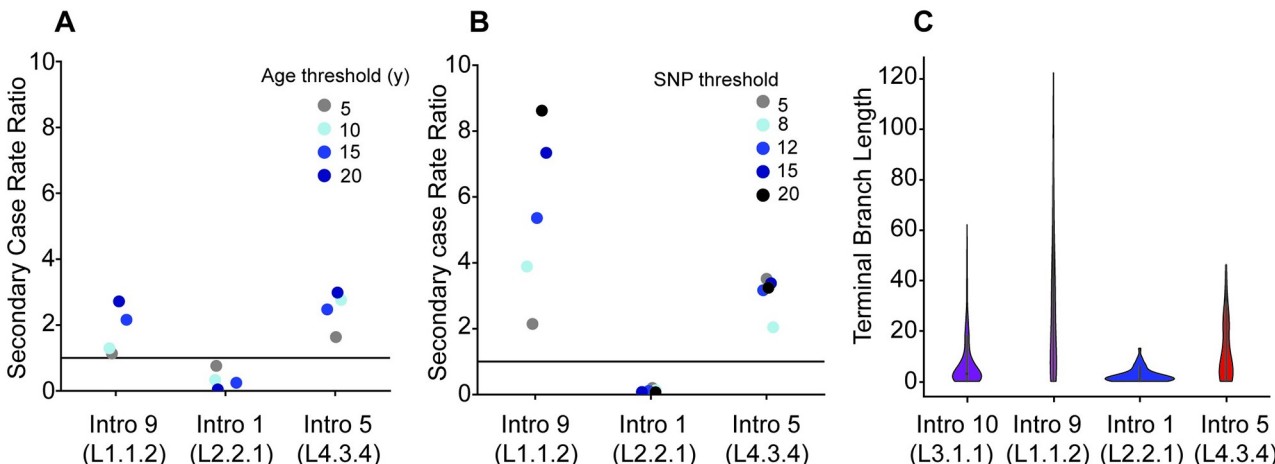

**Fig 4. Transmission analysis using three different approaches.** A and B compare the secondary case rate ratios between Introduction 10 of L3.1.1 and the other successful introductions identified determined based on clustering by using different age thresholds (in years) or different SNP thresholds for A or B, respectively. A secondary case rate ratio of 1 (indicated by a horizontal line) would mean that the secondary case rates of both introductions are the same. C Violin plots comparing the terminal branch lengths between the most successful introductions.

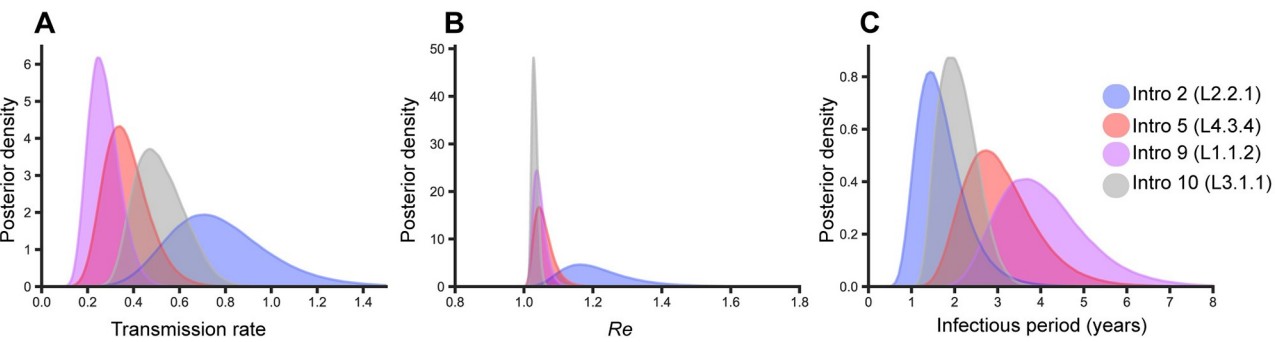

**Fig 5. Transmission analysis of the four most successful introductions using phylodynamic modelling.** A, B, and C compare the posterior distribution of transmission rate, effective reproductive number, and infectious period respectively, for the most successful introductions, as estimated with a phylodynamic birth-death model.

older cluster ages. Accordingly, L3.1.1 and L2.2.1 also had shorter terminal branch lengths, which have also been linked to higher rates of recent transmission (Fig 4C) [30,42].

To allow for potential confounding factors, we carried out a multivariable regression analysis with each of the three proxies for recent transmission as the outcome variable independently (15 years, 5 SNPs, terminal branch length). We found L2.2.1 (Introduction 1) and L3.1.1 (Introduction 10) to be significantly associated with clustering and shorter terminal branch lengths (logistic regression, adjusting for age, sex, HIV status, smoking, and genotype age, p < 0.001, S8 Table).

Finally, to account for potential confounder effects based on genetic distances which result from processes unrelated to transmission and which are extensively discussed in [43], we used phylodynamic modelling to quantify transmission rates ($\lambda$), the effective reproductive number (Re) and the duration of the infectious period for the L3.1.1, L4.3.4, L1.1.2, and L2.2.1 representatives descending from introductions 10, 5, 9, and 1, respectively (Fig 5A-C). While transmission rates represent the average rate at which infected individuals produce new infections (number of transmission events per unit of time), Re represents the total number of transmission events per infected individual during the period the individual remains infectious. We assume that the transmission rate and Re were constant since introduction of the lineage. Consistent with the results obtained from the clustering and terminal branch length analyses, we found that L2.2.1 and L3.1.1 also had the highest transmission rates based on the phylodynamic estimates, followed by L4.3.4 and L1.1.2 (Fig 5A). Despite the notable differences in transmission rates, analysis of L3.1.1, L1.1.2, and L4.3.4 points to similar Re values (Fig 5B), all only slightly over 1, indicating that these populations are experiencing a slow expansion, which is consistent with an endemic status. For the L2.2.1 descendants of Introduction 1, Re indicated a faster expansion (Fig 5 B), which is consistent with the prevalence of this genotype among the ten most common genotypes descending from a common ancestor in Dar Salaam, despite its recent date of introduction (Fig 3). However, the parameter estimates of L2.2.1 were also more uncertain than those of the other genotypes (Fig 5). Re depends on transmission rates and on the period of time during which patients remained infectious (see methods). The differences between the estimates of Re and $\lambda$ can thus be explained by differences in infectious period which was estimated to be longer for L1.1.2 and L4.3.4 strains (Fig 5C).

## Discussion

In this study, we investigated the TB epidemic in Dar es Salaam using a combination of phylogeographical and genomic epidemiological analyses. We found that the MTBC strains

circulating in Dar es Salaam belong to four of the nine main MTBC lineages, with L3 being the most abundant. We found a high genetic diversity for L1, L3, and L4, and by comparing this diversity to a global collection of MTBC genomes, we found that the current TB epidemic in Dar es Salaam stems from several introductions of different MTBC genotypes from diverse regions of the world. Some of these introductions occurred a few centuries ago while others only decades ago. We found that one particular introduction from Central- or South Asia involving L3.1.1, approximately 300 year ago, was by far the most successful, contributing 38.9% of all current TB cases. The epidemiological success of L3.1.1 likely resulted from early introduction combined with an enhanced transmission potential.

Our result that L1, L2, L3, and L4 circulate in the district of Temeke agrees with previous findings [44–46]. This suggests that our sampling in Temeke is a good representation of the MTBC genotypes circulating in the city. With approximately 40% of TB cases caused by L3.1.1, this sublineage is clearly the most successful in Dar es Salaam. L3.1.1, also known as CAS1-Kili (SIT 21) based on the spoligotyping nomenclature [47], circulates also in other East African countries (S8 Fig) [11,48–57], but is most prevalent in Tanzania. In Zambia and Ethiopia, L3.1.1 has been associated with multi-drug resistant (MDR) TB [58,59]. By contrast, we did not detect any MDR isolate among L3.1.1 in our cohort, and the prevalence of drug resistance was generally low, which is in agreement with previous studies from Tanzania [60–62].

Our findings indicate that the current MTBC diversity in Dar es Salaam mainly consists of strains that were introduced directly or indirectly from outside Africa during the last few centuries. Two scenarios could account for this observation: either there was no TB in Tanzania before these introductions, or the original MTBC diversity was replaced by the newly introduced genotypes. According to the first scenario, Tanzania would have been a virgin soil for TB before the introduction of L3.1.1, which is in concordance with old medical reports from colonial times stating that TB was rare before European contact [63]. On the other hand, the currently available evidence points to East Africa as the most likely origin of the MTBC as a whole [3,12,14]. This includes the strong association of *M. canettii* and other so-called smooth mycobacteria closely related to the MTBC, with the Horn of Africa [64], as well as the restricted distribution of the human-adapted MTBC lineages L7, L8, and L9 to East Africa [65–67], and the phylogenetic position of L8 as sister clade of the rest of the MTBC [67]. The decreasing genetic diversity of the MTBC as the distance to East-Africa increases has also been suggested to reflect out-of-Africa migration events of the MTBC [12]. The "Out-of-and-back-to-Africa" hypothesis [12,13] postulates that the MTBC originated in Africa, then spread to the rest of the world, and was subsequently reintroduced to Africa from diverse regions of the world where it could have shifted its optimal virulence in response to the high human population densities of cities in Europe, India and East-Asia [13,14], possibly out-competing less virulent local strains in the process. Our biogeographical and temporal findings are in line with this notion. A replacement by MTBC diversity introduced from Europe has probably happened in South America, where ancient genomes isolated from 1,000 years old human remains have revealed infections with genotypes most closely related to *M. pinnipedii* [68,69], while most human TB cases today are caused by L4 [70]. Generally, such replacements could also explain why dating techniques based on the molecular clock point to a rather young age of the MTBC at around 6,000 years [39,68], while other paleobiologic studies claim that MTBC DNA has been isolated from a bison and humans dated to more than 17,000 and 8,000 years before present, respectively [71,72].

We searched for determinants of the evolutionary success of the different MTBC genotypes sampled in our patient cohort. A possible scenario would be that genotypes that have been introduced earlier would have attained a higher prevalence. While this could explain the dominance of L3.1.1, more generally, the prevalence of the different MTBC genotypes in Dar

es Salaam only partially reflected differences in the timing of their introduction. Local adaptation of MTBC genotypes to the patient population has been proposed to explain the dominance of particular MTBC variants [15,73]. We tested whether MTBC genotypes that co-existed with this host population for longer, and had thus more time to adapt, exhibited differences in virulence and transmission related traits. Virulence, as measured by the degree of harm caused to the host assessed by disease severity parameters at active TB disease stage, did not differ between genotypes. With respect to transmission related traits, we estimated transmission rates for the four most common groups descending from Introductions 10 (L3.1.1), 9 (L1.1.2), 5 (L4.3.4) and 1 (L2.2.1), occurring between approximately 300 and 20 years ago. The oldest introduction (Introduction 10, L3.1.1) and the most recent one (Introduction 1, L2.2.1) showed higher transmission rates per unit of time compared to L1.1.2 and L4.3.4. Yet, the estimated effective reproductive number Re, which gives an indication of the overall transmission averaged over the many bacterial generations since the introduction to Dar es Salaam did not differ much between these different groups. As Re provides a direct inference of overall fitness, these results are consistent with the observation that despite having lower transmission rates, L1.1.2 and L4.3.4 representatives were able to persist in the Dar es Salaam population over time. Our model suggested that patients infected with those genotypes remained infectious for longer periods of time than patients infected with L3.1.1 and L2.2.1. The estimated period of infectiousness could reflect differences in latency periods of these different MTBC genotypes, but could also be affected by differences in sampling proportions linked to potential differences in disease progression. One study in Gambia found that individuals infected with MTBC L6 (also known as *Mycobacterium africanum*) were less likely to progress to active disease compared to individuals infected with other MTBC lineages [74]. In Ethiopia, patients infected with MTBC L7 strains experienced delays in seeking treatment presumably because L7 infections elicited milder TB symptoms [75]. Whether similar differences exist among the MTBC genotypes circulating in Tanzania and elsewhere remains to be explored. While we cannot formally test for local adaption of the dominant MTBC genotypes to the Dar es Salaam host population with the current data, our results revealed that two important conditions for local adaptation to occur were met, namely that there is phenotypic variation in bacterial traits that affect transmission and that this variation is probably, at least in part, genetically determined [76]. Assuming that the strains from L3.1.1 and L1.1.2 that have been introduced into Dar es Salaam around the same time have encountered a similarly susceptible host population upon introduction, our results suggest that different traits affecting transmission in different MTBC genetic backgrounds have evolved, prior or after introduction, and point to bacterial factors as strong determinants of the TB epidemic.

Heterogeneity of the host population could also be invoked to explain the observation that the main MTBC genotypes exhibit different epidemiological parameters, if for example the different MTBC genotypes would transmit preferentially within certain human groups within Dar es Salaam. Patient self-reported ethnicity pointed to a diverse set of ethnic groups, which nevertheless were mostly Bantu. Given the even distribution of MTBC genotypes analyzed across the main ethnicities of our cohort and the high intermingling between different districts within Dar es Salaam, host heterogeneity seems an unlikely explanation for our observations but remains to be formally tested. The immune status of the host can have a strong effect on MTBC trajectories at the patient level, as illustrated by the effect of HIV/TB co-infections, both by increasing TB infection rates and by worsening infection outcomes. We found differences between HIV positive and negative patients, in that the former had atypical lung pathologies, which could result from HIV/TB patients having more disseminated MTBC infections, less restricted to the lungs. However, this aspect is unlikely to explain our observations, as we did not find any association between HIV co-infection and particular MTBC genotypes. One

additional aspect that could further account for the observed differences in the prevalence of the MTBC genotypes is that the founding populations of L3.1.1 might have been larger than those of a similar age, such as L1.1.2, explaining current differences in their prevalence. However, this would not explain the differences we found in transmission rates.

L2.2.1 has previously been associated with increased transmission [30,77,78], often in combination with drug resistance [30,79]. Furthermore, it has been suggested that coevolution is at play with the success of L2.2.1 due to associations with mutations in immune genes [80–82]. In our study though, L2.2.1 did not contain any drug resistance mutations and it has only been introduced very recently, suggesting little time for any coevolution with the Tanzanian population. These findings suggest that inherent strain properties are important for explaining the success of L2.2.1. L3.1.1 had the second highest transmission rate among the four successful groups analyzed. To the best of our knowledge, this is the first report of L3 being a particularly transmissible genotype. Interestingly, in Malawi, sublineage L3.1.1 was found to have increased markedly in prevalence from 1% between 1986–1991 to 13% between 2006–2008 [55]. This observation is consistent with the comparatively elevated transmission rates of L3.1.1 reported here. Generally, L3 has been associated with low transmission [7,42] but in East African countries, also other L3 subgroups than L3.1.1, attain relatively high prevalence contradicting that notion [83,84].

Our study was limited in that the patient recruitment was hospital-based, which could have influenced our sampling. Typically, patients seek care once they feel ill, and it is therefore possible that at that stage of active disease, differences in virulence traits are small. Performing passive hospital-based sampling could also miss subclinical cases, which might still contribute to transmission and thus lead to an underestimate of the prevalence of MTBC genotypes that cause less severe disease. Our observation that patients without MTBC WGS available had a significantly lower chest X-ray score than patients with a bacterial WGS available, possibly reflect such a sampling bias. However, the fact that we found no association between disease severity and MTBC genotype argues against a systematic recruitment bias related to genotype-specific differences in disease severity.

In conclusion, our findings suggest that all MTBC strains causing TB in Dar es Salaam have been introduced from different parts of the world. The four most prevalent genotypes descending from these introductions have different epidemiological characteristics. While L3.1.1 and L2.2.1 exhibited higher transmission rates, L1.1.2 and L4.3.4 have lower transmission rates but persisted in this host population, possibly because they elicit longer periods during which patients might be infectious. These MTBC genotypes have co-existed with the host population of Dar es Salaam for different periods of time, but the duration of this co-existence did not explain the differences in epidemiological characteristics observed. This suggests that different life-history traits have evolved in these different bacterial genotypes, and that the epidemiological characteristics observed are strongly influenced by bacterial factors.

## Methods

### Ethics statement

Ethical approval for the TB-DAR cohort has been obtained from the Ethikkomission Nordwest- und Zentralschweiz (EKNZ UBE-15/42), the Ifakara Health Institute—Institutional Review Board board (IHI/IRB/EXT/No: 24–2020) and the National Institute for Medical Research in Tanzania—Medical Research Coordinating Committee (NIMR/HQ/R.8c/Vol.I/1622). A written informed consent has been obtained from every patient who has been recruited into the TB-DAR cohort.

## Study population

We recruited 1,734 adult sputum smear-positive and GeneXpert-positive patients from a prospective cohort recruited between November 2013 and August 2019 at the Temeke District Hospital in Dar es Salaam, Tanzania (TB-DAR cohort). Sputum samples and detailed clinical and sociodemographic information were obtained for all patients. Tribes indicated are self-reported ethnicities. The bacterial isolates were cultured on Löwenstein-Jensen solid media at the TB laboratory of the Ifakara Health Institute in Bagamoyo. Until 2017, MTBC isolates were shipped to Switzerland for DNA extraction and later DNA was extracted in Bagamoyo and then the DNA shipped to Switzerland for sequencing. Bacterial DNA could be obtained from 1,155 unique patient samples (66%, S1 Fig) while the remaining cultures did not grow. All samples were sequenced with Illumina short-read technology at the Department of Biosystems Science and Engineering of ETH Zurich, Basel (DBSSE). The newly sequenced WGS data can be found under the bioproject PRJEB49562 on ENA.

## Measures of virulence

As a first proxy for virulence, we calculated the TB-score adapted from [85], which is a clinical score consisting of several signs and symptoms such as BMI and fever and that is predictive of mortality [85]. For each of the following symptoms or clinical measures, we assigned a point if present or true: cough, hemoptysis, dyspnea, chest pain, night sweat, anemia, abnormal auscultation, body temperature above 37°C, BMI below 18, BMI below 16, mid upper arm circumference (MUAC) below 220, MUAC below 200. Thus for each TB patient, a maximum of 12 points could be achieved. When categorizing TB-score, values of up to five were considered as mild, values of six and seven as moderate, and above seven as severe. As a second proxy, X-ray-scores were established according to Ralph et al. [86] by two independent senior radiologists at the University Hospital of Basel in all patients with X-rays of sufficient quality (N = 702). The Ralph score is a validated method for grading chest X-ray severity in adult pulmonary TB patients [86]. For categorization of X-ray scores, values below 71 were considered as mild, while the rest was considered as severe according to the optimal cut-off point in the original study [86]. As a third proxy, we determined the bacterial load based on the difference between the first (early cycle threshold) and the last (late cycle threshold) during quantitative PCR (Ct value). The value taken was the lowest out of five probes taken from sputum samples (N = 606).

## Global reference phylogenies for L1-L4

For each of the lineages L1, L2, L3, and L4, we compiled a set of genomes representing the worldwide diversity of that lineage. For L1 and L3, we used the datasets compiled by Menardo et al. [18], which thus far represents the most comprehensive representation of the known geographic range of L1 and L3, consisting of 2,008 and 758 genomes, representing 44 and 32 countries, respectively. We further added 11 and 39 genomes for L1 and L3 sampled in a rural site in Tanzania. To get a good representation of the diversity present within L2 and L4, we gathered previously published genomes and downloaded genomes from public repositories from as many countries as possible. In addition, we newly sequenced 132 and 329 genomes from 16 and 22 countries for L2 and L4, respectively, to increase the representation of African and European L4 and L2 strains (S10 Table). All the genomes selected for the downstream analysis needed to pass our bioinformatic filters, be published at the time of analysis if downloaded, and have a known country of isolation. In addition, the country of patient origin was required for genomes representing samples from European and North American countries.

This selection resulted in 10,103 genomes for L2 and 15,715 genomes for L4, representing 56 and 82 countries, respectively (S7 Fig).

For L4, the 15,715 genomes were separated into four subsets (Africa, Asia, Europe & Oceania, and North- & South America) based on their country of isolation or country of patient origin. A phylogenetic tree was constructed for each of the subsets from an alignment of variable positions using fasttree [87] (options–nt–nocat–nosupport–fastest). Each tree was trimmed with treemmer [88] to reduce the redundancy (option–RTL 0.99), whereby 10 genomes were kept for each country included (-mc 10). A new phylogenetic tree was constructed from an alignment of variable positions of the 6,461 genomes left of the four L4 subsets and again trimmed (option–RTL 0.95, -mc 10), resulting in the final reference set consisting of 4,455 L4 genomes.

For L2, the 10,103 genomes were split into three subsets (Africa, Asian, Others) based on their country of isolation or country of patient origin. The same procedure as for L4 was applied, resulting in a final reference set consisting of 3,505 L2 genomes. The complete list of all WGS included in our study can be found in S10 Table. The newly sequenced WGS data can be found under the bioproject PRJEB50999.

## Whole-genome sequence analysis

The retrieved and newly sequenced FASTQ files were analyzed using the WGS analysis pipeline described in [88]. Briefly, the FASTQ files were processed with Trimmomatic [89] v. 0.33 (SLIDINGWINDOW:5:20) to remove the Illumina adaptors and to trim low quality reads. We only kept reads of at least 20 bp for further analysis. SeqPrep v. 1.2 [90] was used to merge overlapping paired-end reads (overlap size = 15). The resulting reads were then mapped to the reconstructed ancestral sequence of the MTBC [91] using BWA v. 0.7.13 (mem algorithm) [92]. We then marked and excluded duplicated reads with the Mark Duplicates module of Picard v. 2.9.1 [93]. We further performed local realignment of reads around INDELs using the RealignerTargetCreator and IndelRealigner modules of GATK v. 3.4.0 [94]. Reads with an alignment score lower than $((0.93 \times read\_length)—(read\_length \times 4 \times 0.07))$ ($>7$ miss-matches per 100bp) were excluded using Pysam v. 0.9.0 [95]. SAMtools v. 1.2 mpileup [96] and VarScan v. 2.4.1 [97] were then used for SNP calling with the following thresholds: minimum mapping quality of 20, minimum base quality at a position of 20, minimum read depth at a position of 7x and without strand bias. We excluded positions in repetitive regions such as PE, PPE, and PGRS genes or phages, as described previously [15]. The resulting VCF file was then used to create a whole-genome FASTA file. Additional filters were applied as follows: genomes were removed from downstream analysis if they had a sequencing coverage of lower than 30, if they contained SNPs indicative of different MTBC lineages (i.e. mixed infections), if the ratio of variable to fixed variant calls was higher or equal to one, and finally if their number of fixed and variable variant calls was in the lower quartile and in the upper quartile, respectively, of fixed and variable variant call distributions drawn from the complete dataset. We identified lineages and sublineages using the SNP-based classification by Steiner et al. [98], and Coll et al. [47] as well as Freschi et al. [42], respectively.

## Identification of mutations conferring resistance to first-line drugs

All genomes isolated from our cohort were screened for drug resistance mutations as in [99]. Of the mutations found, we identified those affecting rifampicin, isoniazid, pyrazinamide, ethambutol or streptomycin effectivity or a combination of those.

## Phylogenetic analyses and molecular dating

Alignments of variable positions with a percentage of missing data of ≤ 10% were used to construct phylogenetic trees with either FastTree [87] (options–nt–nocat–nosupport–fastest) or RAxML [100] v 8.2.11 with the general time-reversible model of sequence evolution (options -m GTRCAT–V) with a L6 strain as the outgroup (SAMEA5366648). For the reference trees, we accounted for the fact that only variable positions were taken to create the alignment by adjusting the branch lengths accordingly (adjusted branch length = branch length x number of variable positions / number of all positions). To estimate the substitution rate, we selected for each lineage all the samples with known date of isolation from the reference set as well our cohort samples. To test for temporal signal, we performed a date randomization test by running LSD v0.3beta [101] 100 times with randomly shuffled dates of isolation as done previously [18]. All lineages except for L1 passed the date randomization test. We then estimated the substitution rate using LSD for L2, L3, and L4. The substitution rate obtained was used to date the complete dataset including the samples with unknown date of isolation for each lineage. Since L1 did not pass the date randomization test, we took the LSD-based estimate from Menardo et al. [39]. To account for the uncertainties regarding substitution rates, we also included the lowest and highest rate found in the literature for each lineage, if applicable, to provide a range of possible ages. An overview of the substitution rates used to date the trees with LSD [101] can be found in S9 Table. For genomes with an unknown date of isolation, a range was used, consisting of the earliest and latest date of isolation of the set of genomes with known date of isolation for each lineage separately. We thus assumed, for samples with unknown dates, to have been sampled between 1996 and 2018, 1994 and 2019, 1995 and 2018, 1991 and 2019 for L1, L2, L3, and L4, respectively.

## Phylogeographical analysis

To define introduction times to Dar es Salaam of the different MTBC strains, we reconstructed the changes in the ancestral geographical ranges along the tree containing Dar es Salaam genomes as well as the set of genomes representing the worldwide diversity of each lineage. The dated trees described above were used as input into PastML [34] (Maximum likelihood method marginal posterior probabilities approximation (MPPA) plus option forced_joint), in addition to the subcontinental regions of each genome. Tanzania was separated from the remaining East African countries to be able to explicitly look at Tanzania. According to the output of PastML, we identified the introductions of a lineage into Tanzania, extracted the ages of these introductions, and identified the Dar es Salaam genomes resulting from each introduction. Introductions into Tanzania were considered as more successful when they led to at least 12 TB cases in our cohort. For each introduction, we extracted the time since introduction, both as absolute age as well as relative age compared to the age of the MRCA of that lineage. Genomes of strains resulting from an introduction with a relative age of more than 0.2 were considered as early-introduced, while strains resulting from more recent introductions were considered as recently-introduced. A threshold of 0.2 means that at least 20% of a genotype's evolution has occurred in Tanzania. According to the ages of the MRCA estimated with our molecular clock rates, this relative age of 0.2 translates into approximately 159, 205, 189, and 257 years for L1, L2, L3, and L4, respectively. Additionally, we used thresholds of 0.1 and 0.3 to define, whether a strain was early- or recently-introduced in order to make sure our results remained consistent. All visualizations of phylogenetic trees including metadata were done using the R package ggtree [102].

### Sensitivity analysis of geographical and temporal origins

To ensure that the phylogeographic and temporal results were not affected by sampling we down-sampled L1 and L3 10 times independently and performed the phylogeographical and temporal analyses with the down-sampled datasets. For L1 we randomly down-sampled the Asian strains to the number of African strains that were available for L1 (N = 640) and for L3 we randomly down-sampled the African strains to match the number of Asian strains that were available for L3 (N = 293). The final datasets for L1 consisted of 640 Asian samples plus all other non-Asian samples, while the datasets for L3 consisted of 293 African samples plus all other non-African samples.

### Transmission analysis—Clustering

Alignments of variable positions with less than 10% missing data were used to create SNP distance matrices using the Hamming distance (https://git.scicore.unibas.ch/TBRU/tacos). Insertions and deletions were treated as missing data. Transmission was assessed by using three different approaches: 1. Terminal branch length, 2. Clustering based on a SNP threshold, 3. Clustering based on a time threshold. The terminal branch lengths were extracted from the undated phylogenetic trees and multiplied with the length of the alignment of variable positions used to create the trees. To cluster the genomes based on the SNP threshold, the R package cluster [103] with the function agnes and the unweighted pair group average method was used. The thresholds taken as cutoff for patient-to-patient transmission were five, eight, twelve, and fifteen SNPs. For the clustering based on the age threshold, all nodes were extracted from the dated phylogenetic tree where the node was equal or below the threshold and the parent node older than the threshold. Then, all the tip descendants of a node were considered to be in a cluster. The thresholds applied were five, ten, fifteen, and twenty years.

Secondary case rate ratios were calculated as described in [23]. Briefly, for each of the four most successful introductions belonging to L1.1.2 (Introduction 9), L2.2.1 (Introduction 1), L3.1.1 (Introduction 10), and L4.3.4 (Introduction 5), the number of clusters was subtracted from the total number of clustered cases to calculate the number of secondary cases. To account for enhanced transmission opportunities of prevalent genotypes, the number of secondary cases was divided by the number of index cases (number of clusters plus number of unclustered strains) to define a secondary case rate for each successful introduction. We compared the transmission rates between the strains resulting from Introduction 10 (L3.1.1) and the other three successful introductions by calculating secondary case rate ratios for each pair. Thus, we divided the secondary case rate of Introduction 10 (L3.1.1) with each of the other three successful introductions separately to obtain the secondary case rate ratios.

### Transmission analysis—Phylodynamics

Phylodynamic analyses were performed within the Bayesian MCMC framework implemented in BEAST 2 [104]. The variable SNP alignments were augmented with a count of invariant A, C, G and T's to avoid ascertainment bias [105]. A birth-death model was fitted to the alignments for each of the four main introductions (Introduction 10 within L3.1.1, Introduction 9 within L1.1.2, Introduction 5 within L4.3.4, and Introduction 1 within L2.2.2) separately [106]. This model is based on a stochastic birth-death process, with 'birth' events corresponding to transmission events from one host to another (occurring at a rate $\lambda$), while 'death' events occur when a host becomes uninfectious due to recovery or death (occurring at a rate $\delta$). The effective reproductive number $R_e$ was calculated as $\lambda/\delta$. Infected individuals are sampled with sampling proportion *s*, which was set equal to zero before the onset of sampling. During the sampling period, a uniform prior was used for the sampling proportion, with a lower bound

set equal to the proportion of sampled cases in the entire city, and an upper bound set equal to the proportion of sampled cases in the Temeke district only. Upon sampling, infected individuals become uninfectious with probability *r* [107]. Transmission rates, becoming uninfectious rates, migration rates, and sampling proportions were assumed constant through time. A general time-reversible substitution model with gamma-distributed rate heterogeneity (GTR+$\Gamma_4$) was used and a strict molecular clock was assumed. The prior distributions of the model parameters are listed in S11 Table. All model parameters were estimated jointly.

Three independent Markov Chain Monte Carlo chains were run for each analysis, with states sampled every 1,000 steps. Convergence was assessed using Tracer [108]. The percentage of samples discarded as burn-in was set to 10%. The samples after burn-in were pooled together using LogCombiner [104], resulting in at least 250,000,000 iterations in combined chains.

The sensitivity of our phylodynamic inference was assessed by setting less informative prior distributions on the effective reproductive number and becoming uninfectious rate, and by setting two different informative priors on the sampling rate, the first one centered around the district level of sampling and the second one centered around the city level of sampling (S9 Fig and S10 Table). Xml files for the different analyses are provided as extended data (https://github.com/dbrites/TB-DAR-Mtb).

## Statistical analysis

Sociodemographic and clinical characteristics of patients with and without bacterial DNA available were summarized using proportions and compared using chi-squared tests and ANOVA for categorical and continuous variables, respectively. Patient characteristics between MTBC lineages, sublineages, and early and recently-introduced strains, were also summarized using proportions and compared using chi-squared tests for categorical variables and using ANOVA for continuous variables. Self-reported ethnicities are shown for tribes containing at least 70 members among the patients population investigated (either all or only those with a bacterial genome available). Logistic regressions were performed to test for associations between drug resistance and lineage. Adjusting was done for age, sex, HIV status, and smoking. Logistic regressions were further performed to test for associations between the most successful introductions (Introduction 1, 5, 9, and 10) and three transmission measures (5-SNPs threshold, 15 years threshold, terminal branch length). For testing for associations between the most successful introductions and the terminal branch length, a negative binomial regression was applied. Adjusting was done for age, sex, HIV status, smoking, and genotype age. Genotype age represents the minimal amount of time a certain genotype has been circulating among our study population. For strains that were not clustered, this was the time between when the last strain in the study was isolated and the isolation date of the respective strain. For strains that were clustered, genotype age was represented by the age of the earliest isolation date of the respective cluster. Including the genotype age accounted for the fact that genotypes that were introduced longer ago had more time to transmit and thus were more likely to belong to a cluster. The terminal branch lengths between the different introductions were compared using a Kolmogorov-Smirnov test using Python (version 3.7.0). To test for associations between the disease severity measures and early- or late-introduced strains as well as the sublineages, logistic regressions were performed for X-ray and TB-scores and a linear regression for the log10-transformed ct-value representing bacterial load. Adjusting was done for age, sex, smoking, HIV status, and the most common tribes ($> = 70$ members among the patients with a WGS available). Statistical tests were performed in R (version 4.0.3) unless otherwise indicated.

## Supporting information

**S1 Fig. Flow chart illustrative of patient isolates and genomes included in the analysis.**
(PDF)

**S2 Fig. Frequency of A main MTBC lineages and B main MTBC sublineages isolated between 2013 and 2019.**
(PDF)

**S3 Fig. L1 reference tree containing 2161 genomes from 44 countries including 153 Dar es Salaam genomes.** The most important introductions of L1 into Tanzania are marked and the samples from our cohort indicated with a black tippoint. Branches are colored according to the ancestral state estimated with PastML and the pie charts inserted show the marginal probabilities of the ancestral geographical range for the most important introductions as well as the root. The heatmap indicates the sublineages and the bar scale is in years.
(PDF)

**S4 Fig. L2 reference tree containing 3590 genomes from 58 countries including 85 Dar es Salaam genomes.** The most important introduction of L2 into Tanzania is marked and the samples from our cohort indicated with a black tippoint. Branches are colored according to the ancestral state estimated with PastML and the pie charts inserted show the marginal probabilities of the ancestral geographical range for the most important introduction as well as the root. The heatmap indicates the sublineages and the bar scale is in years.
(PDF)

**S5 Fig. L3 reference tree containing 1262 genomes from 33 countries including 504 Dar es Salaam genomes.** The most important introductions of L3 into Tanzania is marked and the samples from our cohort indicated with a black tippoint. Branches are colored according to the ancestral state estimated with PastML and the pie charts inserted show the marginal probabilities of the ancestral geographical range for the most important introductions as well as the root. The heatmap indicates the sublineages and the bar scale is in years.
(PDF)

**S6 Fig. L4 reference tree containing 4795 genomes from 85 countries including 340 Dar es Salaam genomes.** The most important introductions of L4 into Tanzania is marked and the samples from our cohort indicated with a black tippoint. Branches are colored according to the ancestral state estimated with PastML and the pie charts inserted show the marginal probabilities of the ancestral geographical range for the most important introductions as well as the root. The heatmap indicates the sublineages and the bar scale is in years.
(PDF)

**S7 Fig. Countries included in the reference datasets for A L1, B L2, C L3, and D L4.** The numbers in brackets indicate the number of genomes included. The maps were created with the R package rworldmap [109] and the shapefile for the map can be found under the following link: https://www.naturalearthdata.com/http//www.naturalearthdata.com/download/110m/cultural/ne_110m_admin_0_countries.zip.
(PDF)

**S8 Fig. Frequency of L3.1.1 in East African countries found in studies performing molecular typing [11,48–57].** Countries considered as East African were Tanzania, Uganda, Kenya, Rwanda, Burundi, Sudan, Djibouti, Eritrea, Ethiopia, Somalia, Mozambique, Madagascar, Malawi, Zambia, and Zimbabwe. The map was created with the R package

rworldmap [109] and the shapefile for the map can be found under the following link:
https://www.naturalearthdata.com/http//www.naturalearthdata.com/download/110m/
cultural/ne_110m_admin_0_countries.zip.
(PDF)

**S9 Fig. Sensitivity assessment of our phylodynamic inferences** by changing A-C the prior
on the sampling proportion to a Beta(45.1, 954.9) distribution, centered around the district
level of sampling; D-F the prior on the sampling proportion to a Beta(13.7,986.3) distribution,
centered around the city level of sampling; G-I the prior on the effective reproductive number
to a Lognormal(0,1.5) distribution; J-L the prior on the becoming uninfectious rate to a Log-
normal(0,1) distribution.
(PDF)

**S10 Fig. Sensitivity assessment of the geographical and temporal origins by randomly
down-sampling the genomes from Africa to match the number of genomes from Asia.** Rel-
ative (A) and absolute (B, in years) ages of introductions into Tanzania within L3 of the down-
sampled set are shown. Each run represents the results of the analysis of a down-sampled data-
set and the range of the ages of introductions from all the runs are indicated above the point of
the original dataset.
(PDF)

**S11 Fig. Sensitivity assessment of the geographical and temporal origins by randomly
down-sampling the genomes from Asia to match the number of genomes from Africa.** Rel-
ative (A) and absolute (B, in years) ages of introductions into Tanzania within L1 of the down-
sampled set are shown. Each run represents the results of the analysis of a down-sampled data-
set and he range of the ages of introductions from all the runs are indicated above the point of
the original dataset.
(PDF)

**S1 Table. Comparison of clinical and sociodemographic information between patients
with and without bacterial WGS available.** The tribes named are such with at least 70 mem-
bers among our patient population. P-values were calculated using chi-squared tests for cate-
gorical variables and using ANOVA for continuous variables.
(DOCX)

**S2 Table. Comparison of sociodemographic and clinical patient characteristics, for
patients infected with the four main lineages observed using chi-squared tests.** The tribes
named are such with at least 70 members among our patient population with a bacterial
genome available.
(DOCX)

**S3 Table. Drug resistance conferring mutations present in this MTBC population and the
number of genomes observed with the mutation.**
(DOCX)

**S4 Table. Association between drug resistance and lineages.** Logistic regressions were per-
formed and adjusting was done for age, sex, HIV status, and smoking. Odds ratio were calcu-
lated with L1 as baseline.
(DOCX)

**S5 Table. Comparison of patient characteristics and proxies for disease severity between
the most common sublineages.** The tribes named are those with at least 70 members among

our patient population. P-values were calculated using chi-squared tests.
(DOCX)

**S6 Table. Comparison of patient characteristics and disease severity measures between early-introduced and recently-introduced strains.** The tribes named are those with at least 70 members among our patient population with a bacterial genome available. P-values were calculated using chi-squared tests for categorical variables and using ANOVA for continuous variables.
(DOCX)

**S7 Table. Association between disease severity measures and recently- or early-introduced strains.** Logistic regressions were performed for X-ray score and TB-score, while a linear regression was performed for the bacterial load. Adjusting was done for age, sex, HIV status, smoking, and the common tribes. Early-introduced strains were used as baseline to calculate the odds ratio.
(DOCX)

**S8 Table. Association between transmission and main MTBC introductions.** Logistic regressions were performed and adjusting was done for age, sex, HIV status, genotype age (only for the clustering measures 5 SNPs and 15 years), and smoking. Introduction 5 within L4.3.4 was used as baseline. The brackets behind the measures indicate the error distribution and link function used in the generalized linear model.
(DOCX)

**S9 Table. Introductions of MTBC into Tanzania that led to at least 12 cases in our cohort.** Age was estimated using lineage-specific substitution rates inferred from our data and from other publications (see methods for further details).
(DOCX)

**S10 Table. List of WGS included in our study and associated information.** The column TBdar indicates whether this sample was from our cohort.
(TXT)

**S11 Table. Prior distributions for the parameters of the phylodynamic model.**
(DOCX)

## Acknowledgments

Calculations were performed at the sciCORE (http://scicore.unibas.ch/) scientific computing core facility at University of Basel. Genomes were partially obtained from the International Epidemiology Databases to Evaluate AIDS (IeDEA).

## Author Contributions

**Conceptualization:** Michaela Zwyer, Liliana K. Rutaihwa, Lukas Fenner, Jacques Fellay, Damien Portevin, Sebastien Gagneux, Daniela Brites.

**Data curation:** Michaela Zwyer, Liliana K. Rutaihwa, Jerry Hella, Mohamed Sasamalo, Hellen Hiza, Daniela Brites.

**Formal analysis:** Michaela Zwyer, Liliana K. Rutaihwa, Etthel Windels, Fabrizio Menardo, Gregor Sommer, Lena Schmülling, Christoph Stritt, Daniela Brites.

**Funding acquisition:** Jerry Hella, Lukas Fenner, Jacques Fellay, Damien Portevin, Klaus Reither, Tanja Stadler, Sebastien Gagneux, Daniela Brites.

**Investigation:** Michaela Zwyer, Liliana K. Rutaihwa, Etthel Windels, Damien Portevin, Sebastien Gagneux, Daniela Brites.

**Methodology:** Michaela Zwyer, Etthel Windels, Zhi Ming Xu, Daniela Brites.

**Project administration:** Michaela Zwyer, Liliana K. Rutaihwa, Mohamed Sasamalo, Sonia Borrell, Miriam Reinhard, Anna Dötsch, Lukas Fenner, Jacques Fellay, Damien Portevin, Klaus Reither, Sebastien Gagneux.

**Resources:** Jerry Hella, Mohamed Sasamalo, Sonia Borrell, Miriam Reinhard, Anna Dötsch, Hellen Hiza, George Sikalengo, Bouke C. De Jong, Midori Kato-Maeda, Levan Jugheli, Joel D. Ernst, Stefan Niemann, Leila Jeljeli, Marie Ballif, Matthias Egger, Niaina Rakotosamimanana, Dorothy Yeboah-Manu, Prince Asare, Bijaya Malla, Horng Yunn Dou, Nicolas Zetola, Robert J. Wilkinson, Helen Cox, E Jane Carter, Joachim Gnokoro, Marcel Yotebieng, Eduardo Gotuzzo, Alash'le Abimiku, Anchalee Avihingsanon, Damien Portevin, Klaus Reither.

**Supervision:** Lukas Fenner, Tanja Stadler, Sebastien Gagneux, Daniela Brites.

**Visualization:** Daniela Brites.

**Writing – original draft:** Michaela Zwyer, Liliana K. Rutaihwa, Etthel Windels, Christoph Stritt, Sebastien Gagneux, Daniela Brites.

**Writing – review & editing:** Michaela Zwyer, Liliana K. Rutaihwa, Etthel Windels, Jerry Hella, Fabrizio Menardo, Christoph Stritt, Marie Ballif, Robert J. Wilkinson, Zhi Ming Xu, Jacques Fellay, Damien Portevin, Tanja Stadler, Sebastien Gagneux, Daniela Brites.

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
