## [Decision Letter · Decision Letter 0]

9 Nov 2022

Dear PhD Brites,

Thank you very much for submitting your manuscript "Back-to-Africa introductions of Mycobacterium tuberculosis as the main cause of tuberculosis in Dar es Salaam, Tanzania" for consideration at PLOS Pathogens. As with all papers reviewed by the journal, your manuscript was reviewed by members of the editorial board and by several independent reviewers. In light of the reviews (below this email), we would like to invite the resubmission of a significantly-revised version that takes into account the reviewers' comments.

The reviewers had concerns with sampling. If resampling globally and locally is not possible (would be preferable), some form of bootstrapping is necessary to increase confidence in inferences made.

We cannot make any decision about publication until we have seen the revised manuscript and your response to the reviewers' comments. Your revised manuscript is also likely to be sent to reviewers for further evaluation.

Sincerely,

Helena Ingrid Boshoff

Academic Editor

PLOS Pathogens

Michael Otto

Section Editor

PLOS Pathogens

Kasturi Haldar

Editor-in-Chief

PLOS Pathogens

orcid.org/0000-0001-5065-158X

Michael Malim

Editor-in-Chief

PLOS Pathogens

orcid.org/0000-0002-7699-2064

The reviewers had concerns with sampling. If resampling globally and locally is not possible (would be preferable), some form of bootstrapping is necessary to increase confidence in inferences made.

Reviewer's Responses to Questions

**Part I - Summary**

Reviewer #1: The paper "Back-to-Africa introductions of Mycobacterium tuberculosis as the main cause of tuberculosis in Dar es Salaam, Tanzania" by Michaela Zwyer represents an interesting analysis of M. tuberculosis lineages circulating in Tanzania. The main strength of the paper is a framework enabling the authors to investigate and compare strain fates and properties in what could be called natural ecological experiments starting at the time of import to Tanzania of a new strain. A limitation of the study is limiting sampling in Africa beyond Tanzania (sampling is always a pain), which is likely to impact some of the temporal inferences, for some lineages more than others.

I enjoyed the paper and appreciate the effort, but have some concerns, which are detailed below.

Reviewer #2: In their manuscript, the team of Sebastien Gagneux and Daniela Brites iand collaborators descibe the rusults of their analysis of about 1000 genome sequences from Mycobacterium tuberculosis complew strains that were isolated from TB patients in Dar Es Salaam, a large, highly multi-cultural metropole in Tanzania, during 6 years. By this approach the authors identified the presence of a large diversity of genotypes, belonging to the 4 main lineages (L1-L4) of the previously defined M. tuberculosis lineages.

The authors argue that most of these lineages were introduced into that region of East Africa from South or Central Asia and Europe (for L4) about 300 or less years ago. The authors also found that early and recently introduced strains did not seem to differ much in virulence. The authors also suggest that different life-history traits have evolved in these different bacterial genotypes, and that the epidemiological characteristics observed are strongly influenced by bacterial factors.

Reviewer #3: In the present manuscript, Zwyer et al studied the population dynamics of M. tuberculosis in Dar es Salaam (Tanzania) using a cohort of linked tuberculosis index case data and pathogen whole genome sequences. The authors classify strains of lineages into recently versus early introductions and compare transmissibility and virulence properties. One recently (2.2.1) and one early introduction (3.1.1) appear to be linked to increased transmissibility, leading to authors to conclude that while bacterial factors appear important yet incompletely explain prevalence and transmissibility of strains. More than half of the sampled TB epidemic were classified as introduced strains supporting the notion of ‘Out-of-and-back-to-Africa’ hypothesis where Mtb complex emerged in East Africa. The authors conclude that the current epidemic is largely caused by reintroduced strains from elsewhere.

**Part II – Major Issues: Key Experiments Required for Acceptance**

Reviewer #1: 1. Ancestral state mappings can be strongly influenced by sampling density across categories. A prominent example of possibly problematic sampling is L1 (Fig S3), where an Asian origin is inferred. Here it would be interesting to see how the inferences hold up if the Asian samples were randomly downsampled to match the number of African isolates. Biased sampling could well influence the geographic mapping of the branches, and relatively over-sampled regions will more often be inferred as root location because a larger fraction of the diversity in that region has been sampled. These issues are impossible to bypass entirely, but if the state changes (i.e. imports to Tanzania) occur at the same/corresponding nodes also after downsampling the Asian samples, this would lend strength to the analyses. And vice versa. Irrespective of outcome, I believe this should be tested and discussed.

Also, in the pastML analyses, I struggle making sense of the geographical categories: in the methods section (line 706>) it is stated that East African countries were assigned their own country category, but in the figures, only Tanzania has a separate colour. As far as I know, the inferred state changes are linked to the specified locations, so combining them post analyses would be tricky. I would love it if this could be clarified. Also on this note, IF individual African countries were given separate states, I believe this could be problematic as there a re few samples per country in many cases - which would result in very little signal for a transition rate matrix?

2. I think it would be interesting to discuss the inferred origins of the lineages of interest in light of the paper by MB O’Neill et al https://doi.org/10.1111/mec.15120, which formally looks into this for the 6 main lineages. As far as I can see, the match seems to be good.

3. I think it’s interesting that L2 was found to be more transmissible and free of resistance mutations. The authors show L2 in Tanzania to be of relatively recent origin, which suggests limited time for any coevolution with human populations there. I believe this, at least anecdotally, supports a notion of inherent strain differences being more important for transmission, compared to a more hypothetical host compatibility scenario which the authors discuss from approx Line 146. Also, the absence of resistance mutations in L2, even though the numbers are small, support earlier findings https://doi.org/10.1073/pnas.1611283113, that high rates of resistance in L2 is likely explained by societal upheaval and public health collapse than particular lineage traits. I think it would be cool if the authors included a discussion of these aspects in the paper, but this is no requirement on my side.

Reviewer #2: This is an epidemiological investigation based on WGS, thus no further experiments need to be performed. However, the introduction, results and discussion sections are quite repetitive, and the manuscript wold benefit from considerably shortening. Also, there seems to be a bias in the papers that were cited by the authors, indeed most of the discussion is based on hypotheses that were previously published by some of the authors. The paper should be carefully reviesed in the light of the current literature. Finally, while the authors claim that bacterial factors played a large role in the epidemiological characteristics of the TB situation in Dar Es Salaam, they do not provide any further details to this claim. The use of bacterial factors appears very superficial.

Reviewer #3: MAJOR COMMENTS.

Our main concerns with the authors’ approach and the ‘Out-of-and-back-to-Africa’ is sampling bias. Especially as Mtb sequencing efforts have varied substantially across the globe with Africa, with the exception of South African, being the least represented in public genomic data. Furthermore for this study particular biases step from hospital based, as well as the short sampling time that might not allow solid coalescence times and introduction events to be inferred. First, as most TB patients do not get admitted for treatment in hospitals the study population might differ (i.e. be more sick) than the general population of TB patients. Second, introductions were estimated using PastML where the sampled countries used as input will determine what inferences of introductions can be made (using maximum likelihood or parsimonious methods). If countries on the African continent, or regions in Tanzania, were underrepresented such introduction events would be overestimated. Some form of bootstrapping or resampling of the available data both globally and locally from Tanzania is needed to build confidence in the dates of introduction or even in the introduction itself. Or otherwise a loosing of the strength of the claim around introduction of these lineages (L1 and L3) for which there is a strong prior that they are native and continuously spreading in this part of the world for millennia.

I find the relative dating of the introductions as early or late a little arbitrary. It’s the relative age of introduction relative to the MRCA of the lineage, however lineages continuously evolve and in the nomenclature used by the authors may not be sufficiently differentiated from neighboring lineages to warrant the clade/lineage name. How did they define the lineage? is it based solely on the Coll classification and if so the authors should discuss that the naming and classification depended on sampling and data available at the time when the lineage barcodes were developed and this schema did not necessarily use fixation indices. There has been more recent revisions of this classification published 1-2 years ago that have higher resolution and did rely on fixation indices. Overall I think their findings about lineage 2 are clear and consistent with the literature, but those on lineage 1 and 3 require more care. Especially as sub-lineage designations for these lineages were very course/poorly resolved in the Coll et al. schema.

Line 253: It would be important if authors specified the range of isolation dates of the public dataset, as the ability to date introductions using PastML will depend on the dates and diversity of input genomes. Were older pathogen genomes from Tanzania included?

Line 263: Here, authors state that the sampled strains were introduced to Tanzania from different parts of the world. Isn’t an alternative scenario where strains have continued to evolve in Tanzania (ongoing transmission) including continued spread to other global regions not equally likely to explain the observed diversity? See comment above, if no Tanzanian genomes were used to infer introduction events this aspect could be missed.

Discuss how missingness in sampling might be alternative explanations (L.2.2.1 introduced to Tanzania from Asia via West Africa?)

Line 308: About half (42%) did not ‘manage to establish themselves’ – but aren’t they established if they make up half of the epidemic? Where these lineages diagnosed at stable proportions over time?

Line 336: A number of host level factors mediate disease virulence (co-morbidities, diabetes, HIV, age, sex) – it seems these data were available and could thus help build better models of disease severity, especially in a cohort of hospitalized TB patients. Also system level factors are also very important for potentiating disease severity most notably delays in care or inappropriate prior treatment. It is not indicated if host level factors were adjusted for. If this is not a possible a thorough discussion on the limitations of their analysis should be provided, and their results shouldn’t be reported so definitively. I.e. that there is no association between lineage and severity. Only that one couldn’t be made given the lack of full data on the determinants.

Line 500: It appears that time of introduction did not affect virulence and transmissibility properties, seen that both the oldest introduction (L3.1.1) and the most recent introduction (L2.2.1) were linked to increased transmission. Does this invalidate the previous analyses, suggesting that the sampling was inadequate to determine introduction times?

**Part III – Minor Issues: Editorial and Data Presentation Modifications**

Reviewer #1: Line 361 onwards: the discussion of L.3.1.1 having a head start comes across as a bit naïve as little is known about the strains circulating at the time (if any) at the time of L.3.1.1 introduction.

Line 387 “all genomes …” I dont understand this sentence. I mean, “All genomes that ha a common ancestor dated a certain number of years ago…. ” that would include all samples in any tree, no?

Line 392 and Figure 4: the figure, legend and main text doesnt match up, I think. Please make sure that the figure legend is correct for Figure 4. I would also strongly advice to properly label the x axis in panels A and B in a similar fashion as panel C (rather than using numbers 1,2,3 which is a hassle to cross check against the figure legend). Also in Figure 4: The SNP- and age-threshold legends should be sorted numerically which is not the case now.

Line 398: I think the various tree metrics used to assess transmissibility should be discussed in light of the recent paper by Fabrizio Menardo on the topic https://elifesciences.org/articles/76780

Line 469 onwards. Not to be a pedant, but I think it’s a stretch to say that the inferences presented support an “out of and back to Africa” hypothesis. It really only looks at half of that equation, namely introductions to Africa. I believe there is little in this paper supporting an African origin of TB, even though that is a likely scenario based on decades of research.

Line 553: Is it accurate tp say that increased transmission rates for L.3.1.1 were inferred? They are lower than L2 and higher than L4. I miss a benchmark to justify the statement of “increased transmission”.

Reviewer #2: Specific comments:

Line 139 : Here the authors might add additional examples such as PMID: 21408618, etc. PMID: 32019932

Line 141 additional references for African origin of MTBC could be cited e.g. PMID: 23300794, PMID: 25039682 etc

Line 150 may be reference PMID: 30397300 would be useful to add

Line 166 reference PMID: 32019932 could be added here as well

Line 463-465: Original papers on M. canettii could be cited here: PMID: 23291586 PMID: 24520560

Lines L475-478 : it is not clear what the authors want to demonstrate here. The identification of M. pinnipedii-like DNA in 1000-year old mummies suggests a zoonotic transmission whereby people living in the costal regions of South America 1000 years ago had likely contact to sealions and other pinnipeds as hunters. It is unclear how this finding is linked to the presence of the human-to-human spread of M. tuberculosis strains that likely originated from post-Columbian import of M. tuberculosis strains from Europe. From the available data, the word "replacement" reads misleading. There is little evidence that M. pinnipedii-like MTBC infections were a widespread human disease in the South Americas in previous times.

Line 479-483. It is not clear what the authors want to demonstrate with this paragraph. The age estimations range from 70000 Years to 6000. Could be omitted as very speculative.

Reviewer #3: MINOR COMMENTS.

Lines 128-132: the two sentences are somewhat redundant. Recommend revising “Even though TB has been replaced by COVID-19 as the leading cause of death from a single infectious agent, the COVID-19 pandemic has also resulted in an increase of TB deaths (1). While the TB death toll had been decreasing yearly since 2005, it increased again for the first time in 2020, with an estimated 1.5 million deaths (1).”

Also TB has again overtaken COVID-19 as the most deadly infectious disease

Starting at Line 201: Why were only 66% of isolates sequenced, what were the selection criteria. Some of these sentences can be moved to methods

Line 279: Figure 1, it is difficult to distinguish L3.1.1 and L2.2.1 (both orange) in the figure.

Line 342: Seen that the M. tuberculosis complex arguably evolved from East Africa might it be reasonable to assume that the local population is adapted to all lineages?

Please discuss limitation of clustering-based transmission inference (also in light of missing household contact data and a hospital-based cohort).

Line 381: Seen that infection leads to active disease within 1 (max 2) years’ in the majority of case, would lineage specific mutation rates have an impact on 5 or 12 SNP thresholds?

PLOS authors have the option to publish the peer review history of their article (what does this mean?). If published, this will include your full peer review and any attached files.

Reviewer #1: No

Reviewer #2: No

Reviewer #3: No
---

## [Editor Report · Decision Letter 1]

1 Mar 2023

Dear PhD Brites,

We are pleased to inform you that your manuscript 'Back-to-Africa introductions of Mycobacterium tuberculosis as the main cause of tuberculosis in Dar es Salaam, Tanzania' has been provisionally accepted for publication in PLOS Pathogens.

Best regards,

Helena Ingrid Boshoff

Academic Editor

PLOS Pathogens

Michael Otto

Section Editor

PLOS Pathogens

Kasturi Haldar

Editor-in-Chief

PLOS Pathogens

orcid.org/0000-0001-5065-158X

Michael Malim

Editor-in-Chief

PLOS Pathogens

orcid.org/0000-0002-7699-2064

The authors have addressed the reviewers' concerns with the available genome sequences that they had.
---

## [Editor Report · Acceptance letter]

29 Mar 2023

Dear PhD Brites,

We are delighted to inform you that your manuscript, "Back-to-Africa introductions of *Mycobacterium tuberculosis* as the main cause of tuberculosis in Dar es Salaam, Tanzania," has been formally accepted for publication in PLOS Pathogens.

Best regards,

Kasturi Haldar

Editor-in-Chief

PLOS Pathogens

orcid.org/0000-0001-5065-158X

Michael Malim

Editor-in-Chief

PLOS Pathogens

orcid.org/0000-0002-7699-2064